# Modelling the spatial dynamics of oncolytic virotherapy in the presence of virus-resistant tumour cells

**Darshak Kartikey Bhatt**[1,2], **Thijs Janzen**[3], **Toos Daemen**[1], **Franz J. Weissing**[3]*

**1** Department of Medical Microbiology and Infection Prevention, University Medical Center Groningen, University of Groningen, Netherlands, **2** Center for Translational Research in Oncology, Instituto do Câncer do Hospital das Clínicas, da Faculdade de Medicina, da Universidade de São Paulo, São Paulo, Brazil, **3** Groningen Institute for Evolutionary Life Sciences, University of Groningen, Groningen, The Netherlands

* f.j.weissing@rug.nl

**Data Availability Statement:** To improve usability, we provide two different versions of the model for all common operating systems: 1) a terminal-only version, which reads parameters from a configuration file and which can be used to

## Abstract

Oncolytic virotherapy is a promising form of cancer treatment that uses native or genetically engineered viruses to target, infect and kill cancer cells. Unfortunately, this form of therapy is not effective in a substantial proportion of cancer patients, partly due to the occurrence of infection-resistant tumour cells. To shed new light on the mechanisms underlying therapeutic failure and to discover strategies that improve therapeutic efficacy we designed a cell-based model of viral infection. The model allows us to investigate the dynamics of infection-sensitive and infection-resistant cells in tumour tissue in presence of the virus. To reflect the importance of the spatial configuration of the tumour on the efficacy of virotherapy, we compare three variants of the model: two 2D models of a monolayer of tumour cells and a 3D model. In all model variants, we systematically investigate how the therapeutic outcome is affected by the properties of the virus (e.g. the rate of viral spread), the tumour (e.g. production rate of resistant cells, cost of resistance), the healthy stromal cells (e.g. degree of resistance to the virus) and the timing of treatment. We find that various therapeutic outcomes are possible when resistant cancer cells arise at low frequency in the tumour. These outcomes depend in an intricate but predictable way on the death rate of infected cells, where faster death leads to rapid virus clearance and cancer persistence. Our simulations reveal three different causes of therapy failure: rapid clearance of the virus, rapid selection of resistant cancer cells, and a low rate of viral spread due to the presence of infection-resistant healthy cells. Our models suggest that improved therapeutic efficacy can be achieved by sensitizing healthy stromal cells to infection, although this remedy has to be weighed against the toxicity induced in the healthy tissue.

## Author summary

Oncolytic virotherapy is a promising form of cancer treatment that uses viruses to target, infect and kill cancer cells. Unfortunately, this form of therapy is often not effective, due to the occurrence of virus-resistant tumor cells. As it is challenging to assess the

perform demanding simulations on a high-performance computation cluster and 2) a graphical version, where the user is provided with an intuitive graphical user interface and can visually observe the spatial interplay between cell types. The code used for this work and an executable version of the Oncolytic Virus Resistance simulator (OVR) can be found in the Supplementary data or on www.github.com/rugtres/ovr.

**Funding:** The work of FJW is supported by the European Research Council (ERC Advanced Grant No. 789240). D.K.B. received a PhD scholarship from CAPES (Finance Code 001) and ATTP-GSMS (Abel Tasman Talent Program) Groningen. The funders had no role in study design, data collection and analysis, decision to publish, or preparation of the manuscript.

**Competing interests:** The authors have declared that no competing interests exist.

emergence and spread of resistance experimentally or in (pre)clinical studies, we designed a model that allows to study the spatial dynamics of virus-sensitive and virus-resistant tumor cells in various scenarios, and to predict the efficacy of virotherapy. By analysing the model systematically, we demonstrate the importance of 2D and 3D spatial interactions, the effects of viral properties (such as replication rate and range of infection), the properties of virus-resistant cancer cells (such as the cost of resistance), and the sensitivity of healthy (non-tumor) cells towards viral infection. Our goal is to provide a sound conceptual understanding of the mechanisms underlying therapeutic failure, which eventually may lead to the discovery of strategies that improve therapeutic efficacy. We therefore provide the reader with a graphical and a terminal interface of our model (executable on a local computer), allowing practitioners to reflect on their intuition regarding the complex yet fascinating dynamics of oncolytic virotherapy.

## Introduction

Advances in genetic engineering and synthetic biology have allowed the rational design of safer and more efficient anti-cancer biotherapies [1,2]. Oncolytic viruses belong to such a class of therapeutics, where native or genetically modified viruses are employed as agents to preferentially target, infect and kill cancer cells [2,3]. This strategy is based on the rationale that cancer cells often harbour mutations in innate antiviral responses, which makes them a sensitive target of oncolytic virotherapy [4,5]. To this end, a wide range of viral vectors has been tested in clinical trials, suggesting promising therapeutic benefits [3,6]. Unfortunately, only a small proportion of patients treated with oncolytic viruses were found to have a long-term benefit [3,6,7]. Potential explanations could be technical limitations related to virus delivery and treatment, variability in cancer types, generation of weak anti-tumour immunity, individual differences among patients, and occurrence of infection-resistant cancer cells [4,8–10]. Here we focus on the role of resistance, as it remains to be a relatively unexplored area of research [11].

The presence of resistant cancer cells in the tumour tissue may restrict the efficient spread of the virus and thereby undermine its oncolytic potential. The therapeutic efficacy of a virus does therefore not only depend on the properties of the virus but also on the density and spatial configuration of resistant tumour cells, as these factors are crucial for the spatial dynamics of viral spread in the tumour. To unravel the causes of therapeutic failure it is therefore paramount to map the spatial interactions between virus, infection-sensitive cancer cells, and virus-resistant cancer cells, and to investigate the implications of these interactions for tumour eradication. In principle, this can be done with experimental approaches, using in-vitro assays or animal models to investigate the spatial dynamics of the virus and the different cell types present in the tumour. However, technological limitations do not yet allow the large-scale screening that would be required to get a full picture of when, how and why the complex interplay of multiple factors results in therapeutic failure. In a situation like this, mathematical or computational models can be useful, by providing insights into the role and relative importance of the factors governing the spatial dynamics of virotherapy in a tumour and by suggesting adaptations of the therapy that can subsequently be scrutinized experimentally.

Various mathematical and computational models have been developed to elucidate the outcome of virus-tumour interactions, and some have even explicitly considered the geometry of cells and the spatial nature of the tumour tissue in the model design [12–15]. For example, Berg and colleagues [16] developed two- and three-dimensional spatial models that are informed by in vitro experimental data of virus infection in a monolayer of cells (2D) and

tumour spheroids (3D), respectively. They demonstrated that the introduction of a third dimension alters the dynamics of virus-tumour interaction and significantly influences the ability of the virus to eradicate the tumour. Another study considering the spatial dynamics of virus-tumour interactions [17] demonstrated that there is a dichotomy in therapeutic outcomes due to antiviral signalling mediated by induction of interferon production in infected cells. So far, this has been the only modelling study that addresses the possible role of resistance to viral infection. However, to our knowledge, no study has included resistant cancer cells explicitly in a spatial model, thus allowing one to assess the impact of the presence and spatial configuration of such cells on the viral dynamics and efficacy of virotherapy.

We therefore set out to develop such a model. The model considers a tumour tissue with different cell types that proliferate and die at different rates and that are susceptible to viral infection. As it is known that the spatial configuration of the tumour tissue can play a key role in determining the therapeutic efficacy of oncolytic viruses [16], we systematically compare the outcome of three model variants, which consider a regular 2D monolayer, a more natural Voronoi 2D monolayer, and a 3D configuration of tumour cells. In each of these model variants, we aim to understand the interplay of viral and tumour dynamics and, in particular, the factors determining the persistence of resistant cancer cells that underlies therapeutic resistance.

Resistance to virotherapy is a relatively unexplored area. Still, diverse mechanisms have been identified in the literature (reviewed in Bhatt et al. 2021) [11], including interferon-mediated resistance, epigenetic modifications, hypoxia-mediated inhibition, APOBEC-mediated resistance, virus-entry barriers, and spatiotemporal restrictions to viral spread. In view of this diversity, we refrain from modelling a specific resistance mechanism and instead make more generic assumptions on the emergence of resistance.

We also study the role of healthy stromal cells (such as cancer-associated fibroblasts, epithelial cells and endothelial cells) regarding the efficacy of virotherapy as these cells have functional innate immune responses and are resistant to viral infection [9,10,18]. To limit the complexity of our models, we do not consider immune responses mediated by stromal cells, such as antiviral signalling and myeloid or lymphoid anti-tumour responses. In the present study, we focus on the role of stromal cells in the spread of the virus, and we investigate whether and when sensitizing healthy stromal cells can enhance the persistence of the virus and thus improve the therapeutic outcome.

Thus, we systematically assess the impact of virus spread and oncolysis, resistance mediated by cancer and stromal cells, frequency of resistant cancer cells and cost of resistance on therapeutic outcomes. Furthermore, we consider sensitization of stromal cells towards viral infection and the possibility of virus dispersal in the tumour as possible strategies to improve virotherapy. Finally, by comparing the 2D and 3D models we aim to gain insight into the spatial dynamics of virotherapy so that it can be rationally optimized for better outcomes.

## Model description

### Overview of the model

Our model takes the work of Berg *et al.* [16] as a point of departure. We model the growth of tumour and stromal cells on a cell-based spatial grid, using an event-based time structure. Whereas the grid remains static, cell types can proliferate across the grid, reflecting the proliferation of both stromal and cancer cells. Cancer cells can become infected by an oncolytic virus, which is programmed to preferentially target and kill cancer cells while sparing stromal cells. Upon proliferation, tumour cells are able to acquire resistance against the oncolytic virus. Such resistance may come at a cost in that resistant cells may have a lower proliferation rate or

a higher death rate. We explore how this cost of resistance influences the efficacy of the virus. Furthermore, we explore how the susceptibility of stromal cells to accidental infection by the oncolytic virus influences therapeutic outcomes.

## Spatial organization of the tumour

The model consists of a 2-dimensional or a 3-dimensional grid of 'lattice cells' that may either be empty or harbour stromal or cancer cells. In two dimensions, we consider either a regular lattice (Fig 1A), where cells interact with their four direct neighbours (horizontally and vertically), or a lattice resulting from a Voronoi tessellation (Fig 1B). The Voronoi tessellation is obtained by populating the spatial area with $N$ points that are randomly distributed over the area (their coordinates are drawn from a uniform distribution). These points represent the centre of cells; accordingly, $N$ corresponds to the total number of cells in the simulation. Using Fortune's algorithm [19], we calculate the "Voronoi diagram" that specifies the shape and area of each of the $N$ lattice cells: all those points belong to a cell centre that are closer to this centre than to any other cell centre. For the three-dimensional grid, we only considered a regular lattice (Fig 1C), where cells interact with their six direct neighbours (two in each dimension). Extension towards a three-dimensional Voronoi tessellation is straightforward but was not further explored, as it is computationally demanding.

The geometry of the grid thus generated was kept static throughout a simulation and only the properties of the lattice cells were allowed to change (see below). Throughout the manuscript, we used a grid of 10,000 lattice cells, such that in two dimensions, the grid was 100x100 cells (regular and Voronoi grid), while in three dimensions the grid was 22x22x22 cells. S4 Fig shows that, qualitatively, the simulation outcome is only marginally affected by grid size, confirming that our choice of grid size provides representative results. By keeping the total number of lattice cells constant across the grid, we ensure that found differences between grids are due to their spatial organization rather than due to the size of the grid.

## Cell types present in the model

Each lattice cell in the grid can be inhabited by one of four different cell types: healthy stromal cells, uninfected but infection-sensitive cancer cells, infection-resistant cancer cells, and infected (cancer or stromal) cells (Fig 1D). Stromal cells here represent cancer-associated fibroblasts, endothelial cells and epithelial cells, but do not include immune cells. Each lattice cell is either empty or occupied by one of the four cell types.

Uninfected stromal or cancer cells can proliferate (with rates $b_s$ and $b_c$ respectively) and divide into neighbouring empty lattice cells. When multiple neighbouring empty lattice cells are available, one is chosen at random. Virus-infected cells do not divide into empty lattice cells; instead, they can infect inhabited neighbouring lattice cells (with rate $b_i$). If the neighbouring lattice cell is inhabited by a virus-sensitive cancer cell, this cell will be infected with probability 1; if it is inhabited by a virus-resistant cancer cell (see below), the cell will be infected with probability $S_r = 1 - R_r$ (where $R_r$ is the degree of resistance); if it is inhabited by a stromal cell, the cell will be infected with probability $S_s$. In our standard scenario, stromal cells cannot be infected ($S_s = 0$), but we also consider the option that $S_s > 0$. Each cell also has a fixed probability, independent of its neighbours, to die and leave an empty space in the grid (with rates $d_s$, $d_c$ and $d_i$ for stromal, cancer and virus-infected cells respectively; in line with Berg *et al.* [15] we assume that virus-infected cancer cells and virus-infected stromal cells have the same death rate $d_i$). The rates of proliferation and death of each cell type determine the probability of a cell to occupy a neighbouring lattice cell. Cancer cells have a higher rate of proliferation and a lower death rate as compared to normal cells, defining their tumorigenic features. Upon proliferation into an

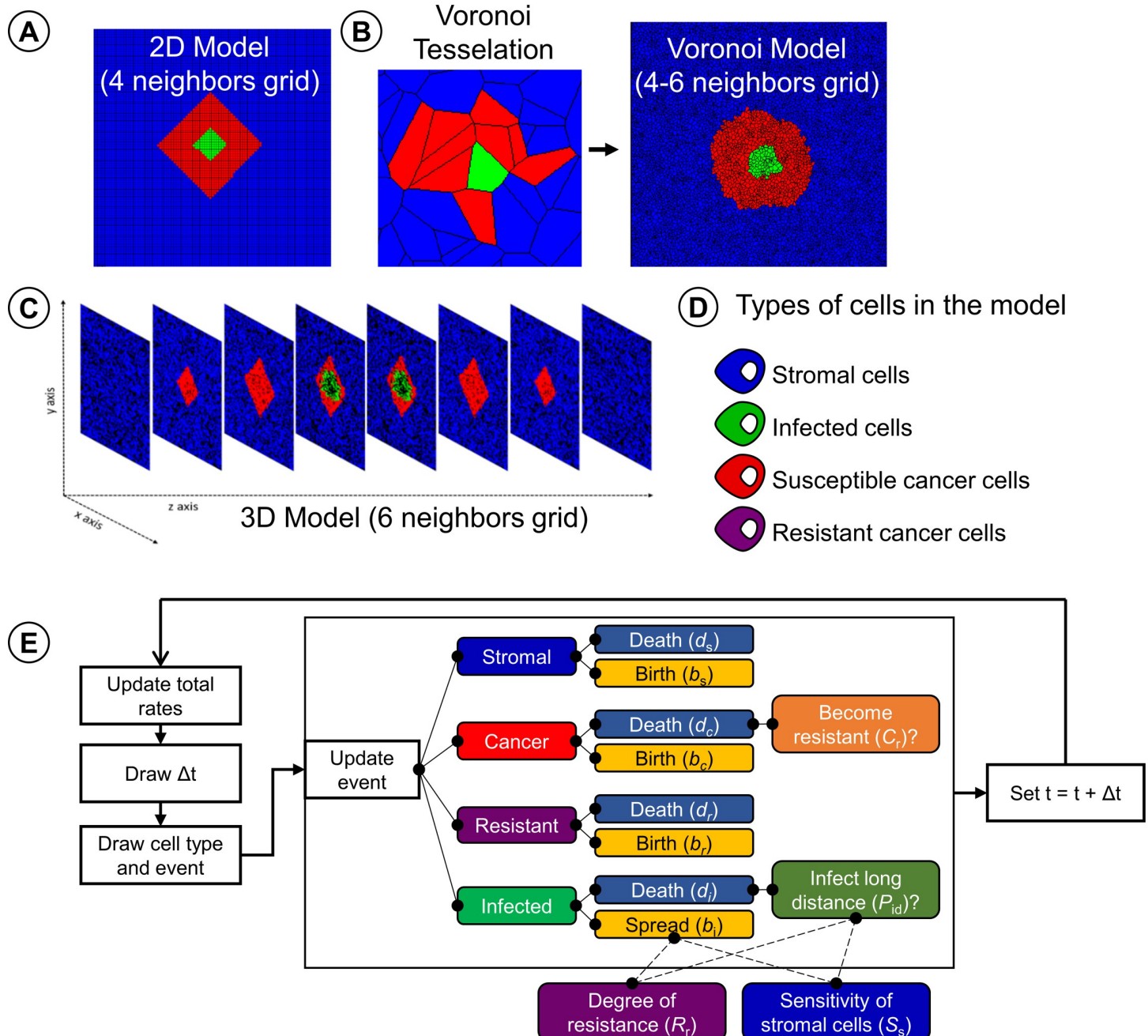

**Fig 1. Visualization of the cell-based model variants.** Spatial organization of (A) the 2-dimensional regular grid model, where each cell has 4 neighbours; (B) the 2-dimensional Voronoi grid model, where each cell has typically 4–6 neighbours; and (C) the 3-dimensional regular grid model, where each cell has 6 neighbours. (D) The four cell types in the model: stromal cells (blue), infection-sensitive cancer cells (red), virus-infected cancer cells (green) and resistant cancer cells (purple). (E) Schematic representation of the event-based model governing the dynamics of virus-cell interactions in the model.

empty cell, the daughter cell of a virus-sensitive cancer cell can, with a small probability $C_r$, acquire resistance to the virus infection. There may be a cost to such resistance, which is reflected by a smaller proliferation rate or a larger death rate for resistant cancer cells ($b_r$ and $d_r$ respectively). By default, we assumed that the degree of resistance is $R_r = 1$, implying that virus-resistant cells cannot be infected at all. Smaller degrees of resistance are considered in (S10 Fig).

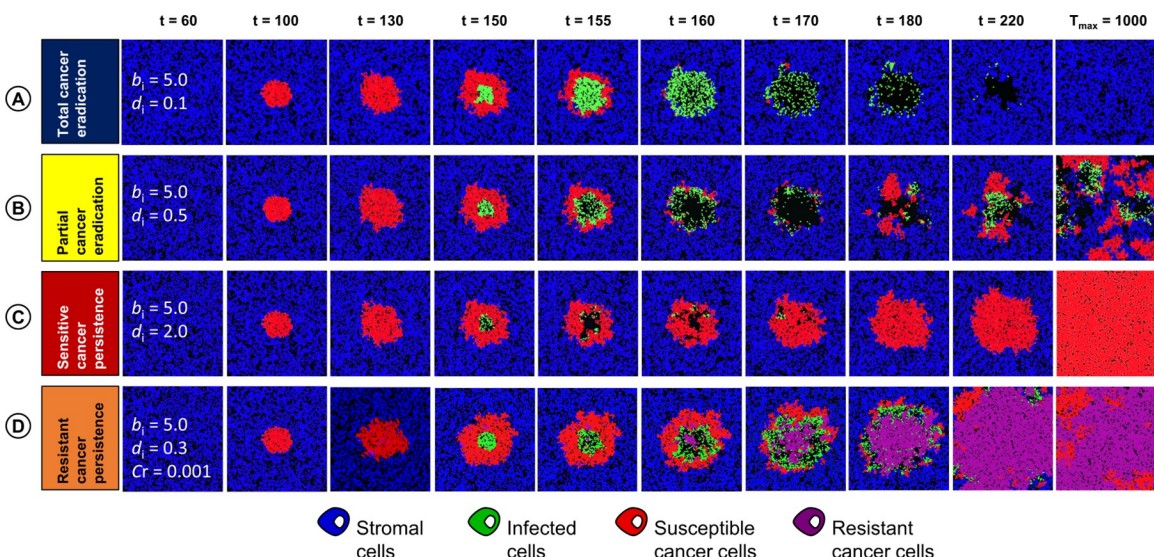

**Fig 2. Visualization of different outcomes of oncolytic virotherapy.** In our model, virus infection in the tumour tissue can result in four therapeutic outcomes: (A) total cancer eradication; (B) partial cancer eradication; (C) persistence of a virus-sensitive cancer; and (D) persistence of a partially virus-resistant cancer. The snapshot ($t = 60$) indicates the parameter values in regard to which the four simulations differ from each other: the rate of viral spread ($b_i$), the death rate of infected cells ($d_i$), and the probability that the daughter cell of a sensitive cancer cell becomes resistant (Cr).

The time structure of the model is event-based, using a Gillespie algorithm [20]. Instead of updating each lattice cell sequentially and incrementing time by a very small amount, the Gillespie algorithm calculates the expected waiting time until a next "event" occurs (e.g. death, proliferation, or infection, see Fig 1E). This has a huge computational benefit, as it is not necessary to update lattice cells that do not undergo an event. To calculate the expected waiting time until the next event, the Gillespie algorithm uses the fact that the waiting time of an event with a fixed probability is exponentially distributed. The expected waiting time until the next event is therefore drawn from an exponential distribution, where the rate parameter of the exponential distribution is the sum of the rates of all possible events in the grid (where a rate is a probability per time unit). Next, the type of event is determined by means of a weighted lottery, where the weights are given by all possible event rates on the lattice. After implementing the event, the event rates on the lattice are updated, the sum of rates is re-calculated, and the time is incremented with the waiting time drawn from the exponential distribution. Some events can have different consequences: the reproduction of a virus-sensitive cancer cell can either lead to a sensitive or a resistant cancer cell, the infection of a stromal cell or a resistant cancer cell can be successful or not, and the death of an infected cell can result in infection via diffusion or not (see below). The likelihood of these consequences is characterized by conditional probabilities (conditional on the occurrence of the event): the probability $C_r$ that the daughter cell of a sensitive cancer cell is resistant, the susceptibility $S_s$ and $S_r$ of stromal cells and resistant cancer cells, and the probability $P_{id}$ that a cell is infected via viral diffusion (see below).

All rates provided are in events per day, such that a rate of 2.0 reflects an event occurring twice per day, with an average waiting time of 0.5 days between events. Simulations were run until $T_{max}$ days had passed, which we chose to be 1,000 days, as at this time point transient patterns had disappeared and firm conclusions could be drawn on the therapeutic outcome (Fig 2).

### Infection via virus diffusion

Apart from the contact-based spread of the virus from an infected cell to a neighbouring susceptible cancer cell (or a stromal cell), the model also considers the possibility of virus diffusion and spread after the death of an infected cell [21,22]. To keep the model simple and computationally efficient, we did not explicitly model the diffusion of virus particles. Instead, we considered the scenario that upon virus-induced lysis a dying cell emits a viral load that can infect nearby cells. In the model, a dying cell infects nearby cells with a certain probability $P_{id}$, where "nearby" means that the distance of the target cell to the dying cell is smaller than the maximal value $D_i$. We consider the direct neighbours of a dying cell to be at distance 1, the neighbours of these neighbours to be at distance 2, the neighbours of these second-order neighbours to be at distance 3, and so on. The infection of neighbouring cells occurs immediately upon death.

### Parameter settings

In view of the huge differences between tumours and types and designs of viral therapies, it is impossible to parameterize our model in a generally accepted manner. We therefore kept the default values of our parameters as similar as possible to those in the study of Berg et al. [16], thus allowing a direct comparison of therapeutic outcomes in the absence (Berg et al. 2019) and presence (our study) of virus-resistant cancer cells. The parameter choice of Berg et al. was partly based on experimental data and partly on their aim to obtain the coexistence of stromal cells, cancer cells, and infected cells in different spatial configurations. In view of the latter consideration, the default parameter setting in our study may not represent the clinically most relevant situation. To compensate for this, we systematically studied a broad range of values for virtually all our model parameters.

### Setup of a simulation

Initially, stromal cells are introduced in the centre of the simulation and allowed to fill the grid until an equilibrium is reached, such that the total number of stromal cells is dynamically stable in time, e.g. the entire grid is populated with the maximum density of normal cells. Cancer cells are introduced in the centre of the grid and allowed to proliferate until time $T_i$, after which the virus is introduced in the centre of the tumour. For all simulations, virotherapy is only carried out once, where a fraction $F_i$ of the cancer cells are infected. To do so, the central tumour cell is determined and infected. Then, all neighbouring cells are infected until a fraction $F_i$ of the cancer cells is infected. The introduction of the virus in the centre of the cancer cells reflects virus injection within the tumour body. We also considered other infection routines (infection in the periphery, infection at random), but as they all gave similar outcomes (see S5 Fig and also observed by Wein et al. [23]), no results on these routines will be reported in this study.

## Results

As various parameters pertaining to the virus, cancer cells and stromal cells influence the dynamics of virus-tumour interaction, we systematically study their influence on the therapeutic efficacy of oncolytic viruses. To this end, we first categorize the result of each simulation into a therapeutic outcome based on the cell types present at the end of the simulation.

### Classification of therapeutic outcomes

Fig 2 visualizes that a simulation can result in four therapeutic outcomes. (A) Total cancer eradication occurs when virus therapy results in the extinction of all tumour cells. In this case,

the virus infection spreads rapidly through the tumour. After the virus-infected cells have died, only stromal cells persist in the end (Fig 2A). (B) Partial cancer eradication occurs when stromal cells, cancer cells and infected cells coexist for a long period of time in a dynamically changing spatial configuration (Fig 2B). In some cases, stromal cells go extinct due to their slower growth rate and only infected and uninfected cancer cells coexist. As infected cancer cells are "handicapped" and do not contribute to tumour formation, therapy is at least partly successful. (C) Persistence of a virus-susceptible cancer occurs when the population of virus-infected cancer cells goes extinct. In the end, the tumour cells take over the population as they outgrow the stromal cells due to their faster growth rate (Fig 2C). This outcome corresponds to therapeutic failure, but as the cancer cells remain virus-sensitive, a new round of treatment might be successful. (D) Persistence of a partially virus-resistant cancer occurs when the virus infection leads to the spread of infection-resistant cancer cells, which in turn results in the extinction of the virus. The resulting population typically consists of a mixture of virus-resistant and virus-sensitive cells (Fig 2D). On a longer-term perspective, the sensitive cells will outcompete the resistant cells (because of their higher growth rate and/or lower death rate), resulting in scenario (C). Still, scenario (D) may be viewed as a more definite failure of therapy, as a new round of treatment will have a low probability of success as long as a substantial fraction of the tumour consists of resistant cells. (E) Finally, there is also the possibility (not shown in Fig 2) that the simulation results in the extinction of all cell types, although we do not consider this as a "therapeutic outcome" of our model.

## Outcome of virotherapy in the absence of resistant cancer cells

Fig 3A illustrates how, in the absence of resistant cancer cells, the therapeutic outcome is determined by the rate of viral spread ($b_i$) and the death rate of infected cells ($d_i$). The outcome largely depends on the ratio $d_i/b_i$ of these rates. Total cancer eradication (Fig 3A, blue area) only occurs if the death rate of infected cells is an order of magnitude smaller than the rate of viral spread (small $d_i/b_i$). In this case, the virus can spread and eliminate all cancer cells. In contrast, a large ratio $d_i/b_i$ results in rapid viral clearance, persistence of the virus-susceptible cancer (Fig 3A, red area) and hence in therapeutic failure. An intermediate ratio $d_i/b_i$ results in partial cancer eradication (Fig 3A, yellow area). Interestingly, there is a small red "wedge" between the yellow and the blue area, indicating that total cancer eradication can also occur for relatively small values of $d_i/b_i$. We will discuss this phenomenon below.

Qualitatively, the dependence of the therapeutic outcome on the ratio of the death rate of infected cells and the rate of viral spread is the same in all three spatial variants of our model. However, in the 3D model partial cancer eradication is observed in a broader parameter range than in the two 2D models (regular 2D and Voronoi). Moreover, the red wedge is smaller in the 3D model. These results are in good agreement with the conclusions drawn from the model developed by Berg et al [16]. Moreover, it is reassuring that our model reproduces crucial aspects of the virus-cancer dynamics (ring-spread or disperse-spread) that was experimentally observed and reproduced by a discrete-time agent-based model by Wodarz et al [14] (S6 Fig).

As an aside, we would like to mention that $b_i/d_i$, the inverse of the ratio $d_i/b_i$, can be interpreted as the expected number of infections caused by an infected cell that is surrounded by susceptible neighbours before its death ($1/b_i$, is the expected lifetime of an infected cell).

## Influence of resistant cancer cells on therapeutic outcomes

The inclusion of virus-resistant cancer cells is a novel aspect of our model. Fig 3B illustrates the effect of such cells on the therapeutic outcome when resistant cancer cells arise (by

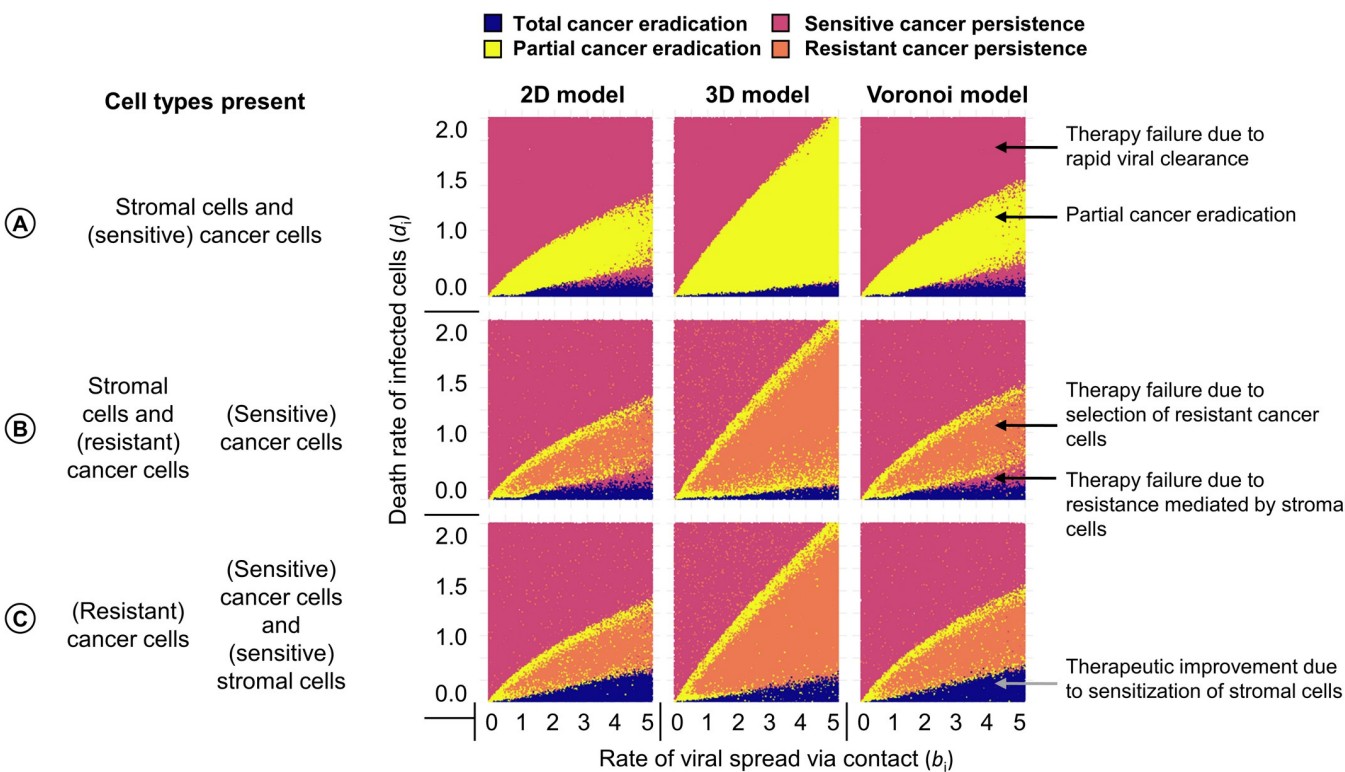

**Fig 3. Effect of the rate of viral spread and death rate of infected cells on the outcome of virotherapy.** For three scenarios: (A) absence of virus-resistant cancer cells; (B) production of resistant cancer cells with probability $10^{-5}$ per cell division; (C) presence of virus-sensitive stromal cells the panels show for three spatial configurations (regular 2D grid; regular 3D grid; 2D Voronoi tessellation) how the therapeutic outcome depends on the rates of viral spread ($b_i$) and death rate of infected cells ($d_i$). Each panel in the figure represents 100,000 simulations: for 100,000 parameter combinations ($b_i$, $d_i$) each simulation outcome is represented by a point, the colour of which indicates the therapeutic outcome. Due to stochasticity, neighbouring parameter combinations can differ in therapeutic outcome. The parameters under investigation were varied, keeping all other parameters at their default values (see Table 1).

mutation) with a frequency of $10^{-5}$ per cancer cell division. In the presence of resistant cells, partial cancer eradication (yellow area) is no longer a prominent outcome. Instead, intermediate ratios $d_i/b_i$ more typically result in the persistence of a partially resistant cancer (orange area) and, hence, in total therapy failure.

Fig 3 shows the therapeutic outcome for a relatively small range of virus-induced spread and death rates of infected cells. S7 Fig presents the outcome of simulations for a much larger range of viral spread rates (0 to 25) and death rate of infected cells (0 to 5). This figure shows that for large infection rates ($b_i > 10$), partial tumour eradication (yellow area) and the persistence of partially resistant cancer (orange area) do no longer occur. Accordingly, the occurrence of resistant cancer cells is of minor therapeutic relevance if the viral spread rate is very large. Additionally, S2 Fig illustrates how the therapeutic outcomes also strongly depend on the birth and death rates of cancer cells, and the resulting $d_c/b_c$ ratio; where a large ratio leads to an increase in the likelihood of total cancer eradication as cancer cells are outcompeted by other cell types.

## Influence of stromal cells on therapeutic outcomes

In the default version of our model, we assume that virotherapy is designed to specifically infect and kill cancer cells, while healthy stromal cells are immune against viral infection. As a consequence, stromal cells may act as a spatial barrier, preventing the efficient spread of the

virus in the tumour. We hypothesized that this barrier effect is responsible for therapeutic failure in the "red wedge" set of parameters in Fig 3A and 3B (the red area between the blue area corresponding to total cancer eradication and the yellow/orange area corresponding to the partial eradication of virus-sensitive cancer cells). To test this hypothesis, we "sensitized" stromal cells, by allowing them to be infected at a similar rate as sensitive cancer cells. Fig 3C shows that, in line with our hypothesis, the "red wedge" is indeed removed by this intervention. In other words, virotherapy with a virus that can infect stromal cells as well as cancer cells can result in total cancer eradication in a situation where the cancer would persist in the case of virus-resistant stromal cells. However, the parameter range where the virus sensitivity of stromal cells is beneficial for the therapeutic outcome is relatively small. Moreover, the susceptibility of stromal cells to viral infection comes at a cost, also in situations where it does not improve the therapeutic outcome. This is shown in S8B Fig: in all scenarios, a smaller population of stromal cells remains when these cells are virus-sensitive. Additionally, when resistant cancer cells can persist, their number is typically positively affected by the presence of virus-sensitive stromal cells, as the presence of these cells increases the selective advantage of resistance (S8C Fig).

Alternatively, S3 Fig illustrates how the therapeutic outcomes are influenced by the birth ($b_s$) and death ($d_s$) rates of stromal cells; where a lower $d_s/b_s$ ratio, increases the likelihood of total cancer eradication, especially if $b_s$ is higher than that of cancer cells ($>1$). This indicates that stromal cells can effectively outcompete cancer cells and prevent the establishment of a tumour.

## Factors influencing persistence of resistant cancer cells

Fig 4A and 4B show how the likelihood of the four therapeutic outcomes is affected by (A) the probability $C_r$ that a virus-sensitive cancer cell acquires resistance due to mutation (see S9 Fig for a more detailed account) and by (B) the degree of resistance of cancer cells to viral infection (see S10 Fig). When the production rate of resistant cancer cells is smaller than $10^{-5}$ (the situation considered in Fig 3B), virotherapy only rarely results in the persistence of (partially) resistant cancer; while this is the typical outcome when $C_r$ is $10^{-3}$ or larger. Meanwhile, a higher degree of resistance to viral infection ($>0.25$) leads to an increasing likelihood of persistence of resistant cancer. However, even a low degree of resistance to viral infection ($<0.25$) is sufficient to result in persistence of resistant cancer when the production rate of resistant cancer cells is high (e.g: $10^{-3}$).

Fig 4C, 4D, 4E show how the "typical" number of virus-resistant cells at the end of the simulation depend on (C) the rate $b_i$ of viral spread, (D) the proliferation rate $b_r$ of infection-resistant cancer cells, and (E) the death rate $d_r$ of resistant cancer cells (all other parameters were kept at their default values, as listed in Table 1 and indicated by a black circle in Fig 4A). The panels show that the conditions for the persistence of resistant cancer cells ($b_r > 0.25$; $d_r < 0.45$) are the same for the three spatial variants of the model (regular 2D, Voronoi 2D, regular 3D). However, the number of persisting resistant cells is much higher in the 3D model than in the two 2D models.

## Tumour progression and time of therapeutic intervention

Considering that tumour progression in time leads to an advanced stage of cancer and often accumulation of resistant cancer cells in the tissue, we assessed how the timing of virotherapy affects the therapeutic outcome. Fig 5A shows how the outcome depends on the time of virus introduction in the tumour ($T_i$) in the 2D Voronoi model (see S11 Fig for the regular 2D and the 3D model). In fact, there is not *one* outcome per introduction time $T_i$ of the virus: replicate

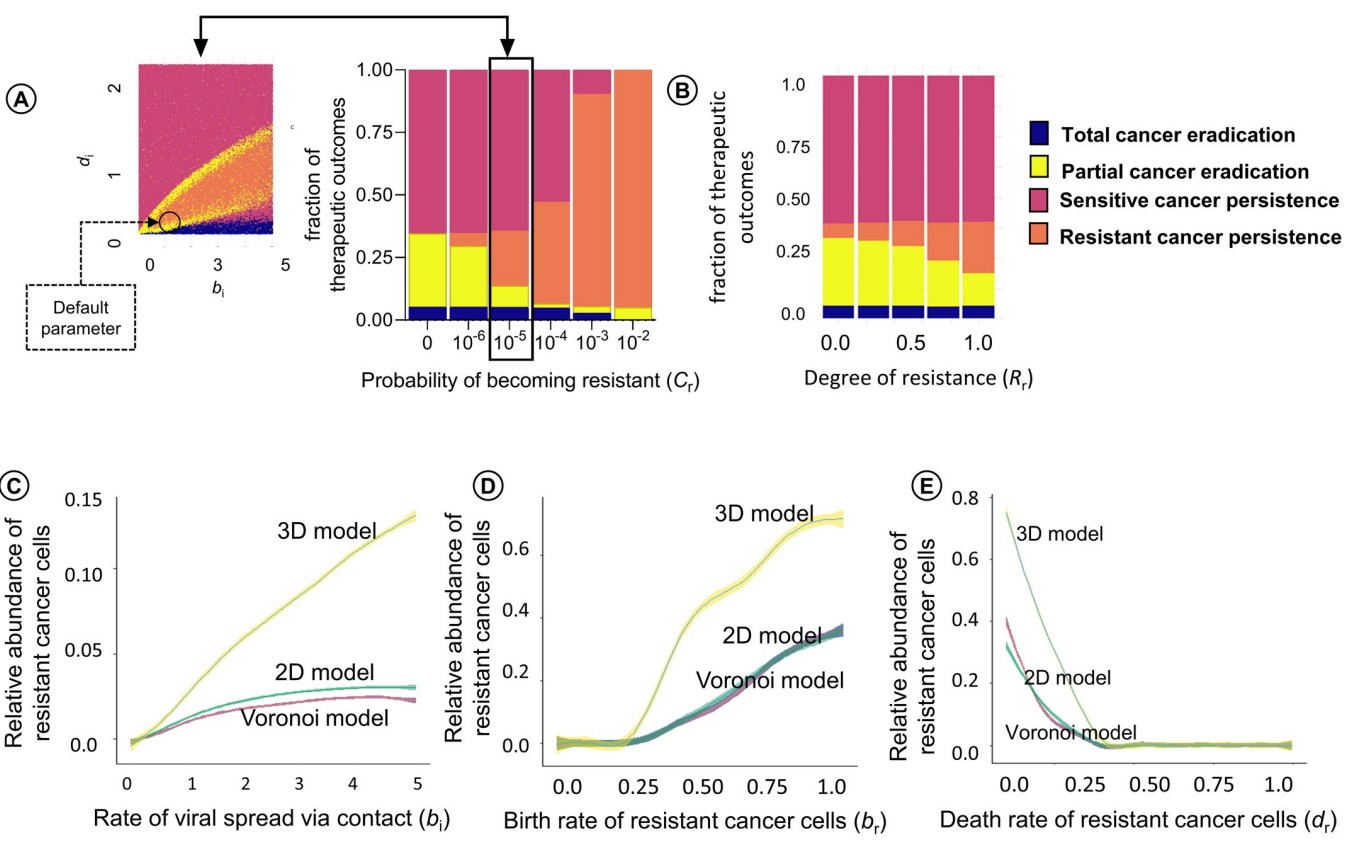

**Fig 4. Factors influencing the persistence of resistant cancer cells in the tumour.** (A) Effect of the probability of becoming resistant ($C_r$) and the degree of resistance ($R_r$) on the therapeutic outcome in the Voronoi model. For the same range of parameter values as in Fig 3 (rate of viral spread ($b_i$) and death rate ($d_i$) of infected cells), 100,000 simulations (for A) and 10,000 simulations (for B) were run and classified as to their therapeutic outcome. The bar chart indicates the likelihood of the four outcomes for six values of $C_r$ and five values of ($R_r$). For ease of interpretation, the production rate of $10^{-5}$ per cell division is highlighted and linked to the Voronoi variant of Fig 3B. The size of the four bars in the bar chart is proportional to the areas indicated by blue, yellow, red, and orange in the panel to the left. Effects of (C) the rate $b_i$ of viral spread, (D) the proliferation rate $b_r$ of infection-resistant cancer cells, and (E) the death rate $d_r$ of resistant cancer cells on the frequency of resistant cancer cells at the end of a simulation run. For graphs C to D, 10,000 simulations were run (for 1,000 parameter configurations and 10 replicates per configuration) for each of the three spatial configurations (2D grid, Voronoi, 3D grid), varying the parameter under investigation and keeping all other parameters at their default values (indicated by a black circle in Fig 4A). The coloured lines indicate the mean per parameter value, and the coloured envelopes around the lines correspond to the 95% confidence band.

simulations for a given introduction time differ in therapeutic outcome, and Fig 5 represents the frequency distribution of these outcomes (for different introduction times, all other parameters were kept at their default values). When stromal cells are resistant to infection (top panel), the distribution of outcomes differs considerably depending on whether virotherapy starts before or after day 250. When the virus is introduced before day 250, total cancer eradication is only rarely achieved. Still, it is advantageous to start therapy as early as possible, as this maximizes the probability of partial cancer eradication. Interestingly, total cancer eradication is achieved in about 10% of the cases when virotherapy is started after 250 days. This can be explained by the fact that most stromal cells have disappeared when virotherapy starts at a late stage (S11B Fig). As shown above, infection-resistant stromal cells may act as a spatial barrier, preventing the efficient spread of the virus in the tumour. It can therefore be beneficial to start virotherapy at a time when this barrier has disappeared (due to the fact that the stromal cells were outcompeted by the cancer cells). However, a late start of virotherapy has the downside that the persistence of a partially resistant cancer is the most likely outcome (Fig 5) and

**Table 1. Model parameters, their default values, and the range of values investigated in the simulations.** Parameter values marked by a star are adapted from Berg et al. [15].

| Parameter | Explanation | Default value | Range |
|---|---|---|---|
| grid size | Number of lattice cells | 10,000 | 500 to 250,000 |
| $b_s$ | Proliferation (birth) rate of healthy stromal cells [unit: events per cell per day] | 0.5* | - |
| $d_s$ | Death rate of healthy stromal cells [events per cell per day] | 0.2* | - |
| $b_c$ | Proliferation (birth) rate of infection-susceptible cancer cells [events per cell per day] | 1.0* | - |
| $C_r$ | Probability of acquiring resistance per cancer cell division | $10^{-5}$ | $10^{-6}$ to $10^{-2}$, and 0 |
| $d_c$ | Death rate of infection-susceptible cancer cells [events per cell per day] | 0.1* | - |
| $b_r$ | Proliferation (birth) rate of infection-resistant cancer cells [events per cell per day] | 0.9 | 0 to 1 |
| $d_r$ | Death rate of infection-resistant cancer cells [events per cell per day] | 0.2 | 0 to 1 |
| $b_i$ | Rate of viral spread via contact with susceptible cells [events per cell per day] | 1.2* | 0 to 25 |
| $d_i$ | Death rate of infected cells [events per cell per day] | 0.1* | 0 to 5 |
| $S_s$ | Susceptibility of stromal cells to viral infection [probability that an infection attempt is successful] | 0 | 0 to 1 |
| $S_r$ | Susceptibility of resistant cancer cells to viral infection [probability that an infection attempt is successful] | 0 | 0 to 1 |
| $R_r$ | Degree of resistance of a resistant cancer cell [probability, $R_r = 1-S_r$] | 1 | 0 to 1 |
| $P_{id}$ | Probability of "infection at a distance" [probability that a cell is infected upon virus-induced lysis of a nearby cell] | 0* | 0 to 1 |
| $D_i$ | Distance from a focal cell in terms of lattice cell layers | 0* | 0 to 3 |
| $F_i$ | Fraction of cells initially infected | 0.1* | - |
| $T_i$ | Time of virus introduction [days] after the start of tumour growth | 50 | 0 to 400 |
| $T_{max}$ | Total runtime of the simulation [days] | 1,000* | - |

the number of resistant cancer cells is maximized (S11C Fig). These negative effects of a late-starting virotherapy are more pronounced in the 3D variant of the model than in the two 2D variants (S11C Fig). Importantly, the effect of the time of treatment is stronger for other parameter combinations, most notably those resulting in partial cancer eradication and those resulting in the persistence of sensitive cancer due to resistance mediated by stromal cells (S12 Fig).

## Dispersal of virus in the tumour

Up to now, we only considered contact-based transmission of the virus. If viral transmission only occurs between neighbouring cells, it can be undermined by a barrier of virus-resistant cells (e.g. stromal cells). Fig 6 shows how the therapeutic outcome is affected if the virus cannot only spread via cell-to-cell contact but also via diffusion over small distances. The four panels in Fig 6B show how, for our default parameters (indicated by a black circle in Fig 4A) and the dispersal distances 0, 1, 2 and 3 (illustrated in Fig 6A), the likelihood of the various therapeutic outcomes changes with an increase in $P_{id}$, the probability that upon the death of an infected cell a given cell in the diffusion neighbourhood of the dying cell is infected. The left-most panel in Fig 6B (dispersal distance zero) considers the case where infection via dispersal does not occur and all virus transmission happens via cell-to-cell contact. As we have seen before, there are two therapeutic outcomes: partial cancer eradication (in about 75% of the simulations) and the persistence of a partially virus-resistant cancer (about 25%). Obviously, the outcome does not depend on $P_{id}$ (as the diffusion neighbourhood is empty). The three other panels in Fig 6B illustrate how strongly the therapeutic outcome can be improved if viral diffusion is a potential form of viral transmission: already a dispersal distance of 2 or 3, virtually all simulations resulted in total cancer eradication; and even for a dispersal distance of 1, total cancer eradication was the most likely outcome for large values of $P_{id}$.

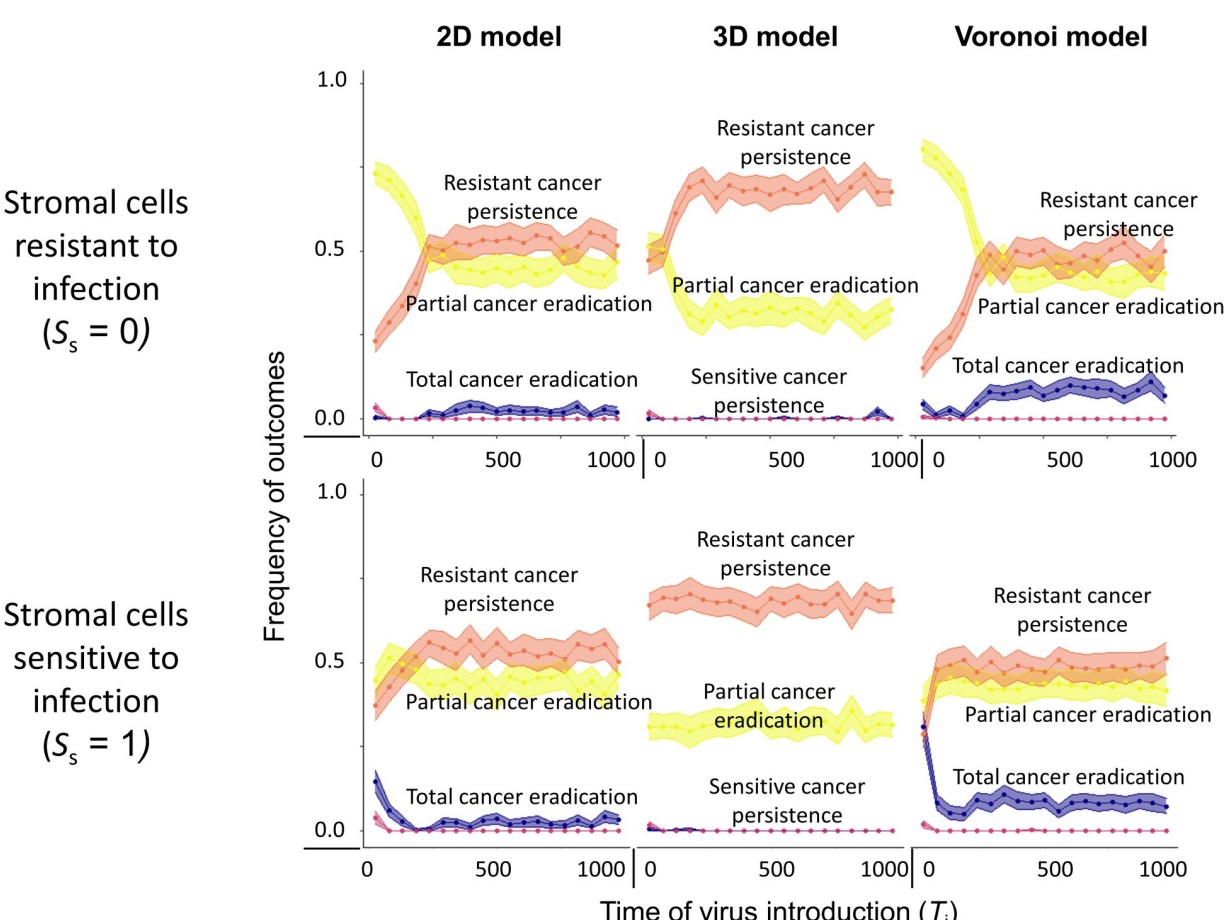

**Fig 5. Effect of time of treatment on therapeutic outcome.** Effect of the start of virotherapy ($T_i$) on the therapeutic outcome in the Voronoi, 2D regular and 3D regular model. The top row describes the situation ($S_s = 0$) when stromal cells are resistant to viral infection, whereas the bottom row describes the scenario ($S_s = 1$) when stromal cells are sensitive to viral infection. Each panel represents 10,000 simulations per spatial configuration: for 1000 virus introduction times $T_i$ 10 replicate simulations were run, keeping all other model parameters at their default values (see Table 1). Due to stochasticity, the replicates for a given introduction time could differ in therapeutic outcome at time 1000. Coloured lines indicate the mean value, coloured envelopes indicate the 95% confidence band. Mean and confidence intervals in (A) were obtained via nonparametric bootstrapping.

Alternatively, Fig 6C illustrates the influence of $P_{id}$ and dispersal distance (1 and 2) on the therapeutic outcomes for a range of rates of viral spread via contact and death rate of infected cells. As expected, higher values of $P_{id}$ improve therapeutic outcomes even for a dispersal distance of 1. Interestingly, parameter values of a large $d_i/b_i$ ratio also result in complete tumour eradication (in contrast to Fig 3) when we consider infection through viral diffusion. There is also a shift in the outcomes leading to partial cancer eradication or persistence of resistant cancer towards the y-axis in the panels; indicating that higher death rates of infected cells are associated with a better spread of the virus (partial cancer eradication instead of sensitive cancer persistence) but also selects for the persistence of resistant cancer cells.

## Discussion

The processes underlying therapeutic resistance to oncolytic viruses are so intricate that intuition and verbal reasoning can provide an incomplete and misleading account. Through a modelling approach, our study provides insight into the dynamics of oncolytic virotherapy

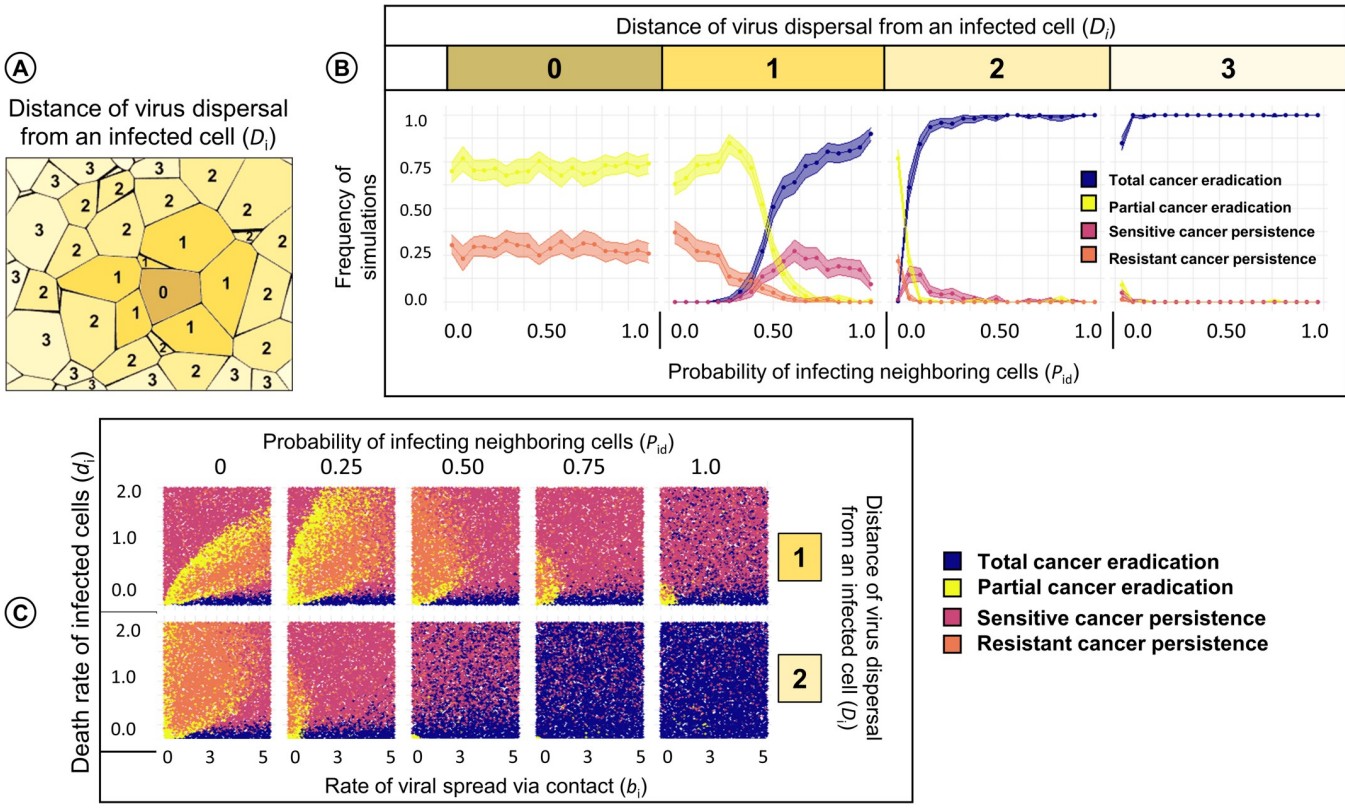

**Fig 6. Effect of viral dispersal distance and the probability that neighbouring cells are infected via diffusing viruses on the therapeutic outcome.** In the Voronoi model, we considered various dispersal distances of the virus. For each dispersal distance, we changed the probability $P_{id}$ that upon the death of an infected cell the cells in the diffusion neighbourhood of that cell are infected, keeping all other parameters at their default values. (A) Interpretation of the viral dispersal distance, here ranging from 0 to 3. A focal cell marked "0" has distance zero to itself, distance one to its neighbours, distance two to its neighbours, etc. (B) Therapeutic outcome for dispersal distances 0, 1, 2, and 3 in relation to the probability $P_{id}$ of being infected by an infected neighbour. For each graph, 10,000 simulations were run; coloured lines and envelopes indicate the bootstrapped mean simulation outcome and the 95% confidence band. In (C) we changed the $P_{id}$ and dispersal distance for the same range of parameter values as in Fig 3 (rate of viral spread and death rate of infected cells) and 10,000 simulations were run for each graph. Each simulation outcome is represented by a point, the colour of which indicates the therapeutic outcome.

and resistance within a spatial framework. Our results demonstrate that the outcome of virotherapy depends not only on the parameters governing virus replication and the spatial architecture of the tumour but also on the presence of resistant cancer and stromal cells that act as a barrier to the spread of oncolytic virus. More importantly, our results also provide an idea of how therapeutic outcomes can be improved by enhancing virus dispersal in the tumour and by sensitizing stromal cells towards virus infection. Overall, our model provides a systematic understanding of the relative effects of various model parameters on the therapeutic outcomes of oncolytic virotherapy. For ease of access, we provide executables of the model (see *data availability statement*), allowing users to challenge their insight and intuition by conducting a spectrum of *in silico* experiments themselves.

Our model is mainly intended as a conceptual tool that elucidates the intricate interplay of processes relevant to oncolytic virotherapy. Accordingly, we kept the model assumptions as simple as possible. Yet, we think that our results and conclusions are not unrealistic, as our model set-up and choice of parameters follow the experimentally validated study of Berg et al. [16]. It is reassuring that our model outcomes also align well with the experimentally validated study of Wodarz et al. [14], despite considerable differences in model structure (see S6 Fig for details). In agreement with the studies of Berg et al. and Wodarz et al., we found that, for a given set of parameters, the therapeutic outcome is not deterministic but stochastic. For

example, when we study the influence of the rate of viral spread and the death rate of infected cells (Fig 3), we find that different outcomes are possible for neighbouring sets of parameters. This may partly explain the variation in clinical outcomes. Stochastic effects are also evident in Fig 5, where for the same time of virus introduction, various outcomes are possible, ranging from complete tumour eradication to the persistence of resistant cancer.

This is the first theoretical study that explicitly addresses the implications of resistance against viral infection for oncolytic viral therapy. Our results suggest that even when resistance arises at a low frequency, such as at a rate of $10^{-5}$ per cell division, resistant cancer cells limit the viral spread and prevent tumour eradication (Fig 4A and S9 Fig). We also looked at rates ranging from $10^{-6}$ to $10^{-2}$, as not much information is available about the true occurrence of infection-resistant cells in the tumour. However, this range seems reasonable as we find at least 1–10% infection-resistant cells in patient-derived cancer cell lines [8,24]. We assume that resistance to viral infection may come at a metabolic or physiological cost that reduces the rate of proliferation or increases the death rate of resistant cancer cells. This is justified by the experimental observations that the production of antiviral factors and related signalling pathways can cause the arrest of cell-cycle, thereby decreasing the proliferation rate [25], and may also lead to stress-induced cell death [26,27]. Not surprisingly, our simulations indicate that a high cost of resistance is associated with poor survival of resistant cancer cells and leads to a significant decrease in their frequency (Fig 4D–4E). The cost of resistance has a significant influence only in scenarios where the tumour persists due to the selection of resistant cancer cells. Perhaps more surprisingly, our results indicate that in the scenario with a faster rate of viral spread there is rapid clearance of infection-sensitive cancer cells, which in turn promotes the selection and survival of resistant cancer cells. Indeed, this has also been found to be true in practice where therapeutic intervention causes the selection of resistant clones [28].

Our study highlights the importance of stromal cells for oncolytic virotherapy. Even if stromal cells are not affected by the presence of the virus (as we assume in most of our simulations), they can prevent viral spread as well. Hereby they can contribute to therapy failure when viral infection and viral replication are specific to cancer cells only. Our observations from the simulations are in line with various experimental studies that observe an improvement in therapeutic outcomes due to either low stromal density [29], or high density of target cancer cells [30], or upon sensitizing normal cells, or upon using viruses that are not only specific for cancer cells [31–33]. We observe that sensitizing stromal cells to viral infection increases the likelihood of the virus to spread from an infected cell to non-infected (sensitive) cancer cells in the neighbourhood, which results in an improvement in the efficacy of tumour eradication. However, sensitizing stromal cells does not affect the persistence of resistant cancer cells and comes at a cost of stromal cell death and possible cytotoxicity to the healthy tissue present in the microenvironment (S8 Fig). Moreover, cell-to-cell transmission of virus during early stage of cancer growth is prevented by the presence of resistant stromal cells, whereas during later stages is due to resistant cancer cells (Fig 5). The reader should notice that, for the sake of simplicity, our model assumes that infected stromal cells have the same rate of spreading the virus and the same death rate as infected cancer cells. This assumption will often not be realistic. However, a difference in the rate of virus spread (via contact) between the two types of cells can easily be achieved by changing the susceptibility of stromal cells to viral infection. When studying differences in the death rates of infected stromal and cancer cells (in an extended version of the model), we observed some potentially interesting patterns, which we will study in detail in future work.

However, not all oncolytic viruses depend on a cell-to-cell mode of transmission and instead, can infect cells by diffusing through extracellular space. We did not include viral diffusion in our model, as it would have slowed down the simulations considerably. However, we

included 'infection at a distance', which gives an indication of the effects of diffusion. Here, we find that extending the infection distance even by a single degree of neighbour can significantly increase the number of outcomes that result in partial or complete tumour eradication. Our findings are in agreement with the observations that an improved diffusion coefficient increases the number of infected cells in the vicinity [34], and that the probability of infection can depend on neutralization by antibodies and frequency of infectious virus particles released upon cell lysis [3,6,35]. Again, these results highlight the importance of explicitly considering spatial features and extracellular factors in order to gain insight in understanding virus-tumour interactions.

Our model addressed the importance of the size and spatial configuration of the tumour. We find that grid size (total cell number) has a clear impact on the therapeutic outcome (S4 Fig): in general, a larger grid size is favourable for the persistence of resistant cancer. This is explained by the fact that, due to the larger number of cells, the emergence of resistant cancer cells is more likely, even if they arise at a low frequency ($10^5$) per cell division. Nevertheless, the overall results are robust and comparable between grids of different sizes, provided that they contain at least 10,000 lattice cells. Considering the spatial configuration of the simulation grid, we find that in 2D, both the Voronoi (three to six neighbours) and the regular model (four neighbours) yielded very similar results, despite the difference in connectivity between cells. This is in agreement with the conclusions of Berg et al. [16], who also found strong similarities between regular and Voronoi grids. However, when including a third dimension, we find that (sensitive or resistant) cancer cells were able to persist for a broader range of parameters. Such differences between 2D and 3D settings could hint at potential discrepancies when comparing *in vitro* results obtained in a (more or less) two-dimensional petri-dish environment, and *in vivo* results obtained in a three-dimensional setting [16,36,37]. One might hypothesize that the differences in outcomes reflects the difference in the number of boundary cells in the 2D model (5% of the total cells) and the 3D model (25% of the total cells) in a grid of 10,000 lattice cells. However, this is unlikely because we do not find any differences in outcomes upon changing the grid size of the 2D or 3D models (S4 Fig), while the fraction of boundary lattice cells decreases strongly with grid size. Alternatively, the differences between the 2D and 3D models might be caused by the differences in the number of neighbour cells. Again, this is unlikely as the two 2D models yield very similar results despite a clear difference in the average number of neighbours per lattice cell (4 in case of a regular grid and 4–6 in the case of a Voronoi grid). On the other hand, the Voronoi and the 3D model have a similar number of neighbouring cells but do differ in the outcomes. We therefore hypothesize that the differences in simulation outcome are not caused by neighbour effects but by the fact that the degree of stochasticity is higher in the 3D model because of the extra degree of freedom.

To minimize the complexity of our model, we have neglected intra-tumoral heterogeneity. The model could be extended by introducing different cancer cell clones that vary in their rates of proliferation and death and susceptibility to viral infection. Additionally, one may also implement a degree of susceptibility to viral infection and resulting changes in the rate of cell death to have a finer understanding of the virus-cancer dynamics. Perhaps most importantly, we have neglected the effect of immune responses during virotherapy. This will be the subject of a future study where we aim to model the anti-tumour and anti-viral immune responses in a similar spatial configuration.

Current experimental research has demonstrated the possibility of employing effective oncolytic virotherapy candidates in clinics; however, therapeutic resistance will remain to be a challenge unless efforts are made to understand the underlying mechanisms. We hope that our computational approach aids in defining the impact of the various factors that may influence resistance and thereby therapeutic efficacy of virotherapy. We are confident that our results,

along with experimental observations, can assist the scientific community in improving the design of virotherapy.

## Supporting information

**S1 Fig. Comparing the visualization of the Voronoi model to the 3D model.** A snapshot of the (A) Voronoi model and the (B) 3D model at a runtime (t = 155) was taken for comparison of the model dynamics. The grid size was set at $100^2$ for the Voronoi model and at $22^3$ for the 3D model to have a comparable (~10,000) number of total lattice cells. The parameter values of rate of viral spread ($b_i$), death rate of infected cells ($d_i$) and probability of becoming resistant ($C_r$) is indicated in the figure.
(TIF)

**S2 Fig. Effect of birth and death rate of cancer cells on the outcome of virotherapy.** Therapeutic outcomes in relation to the birth ($b_c$) and death ($d_c$) rates of cancer cells for the Voronoi model were considered by keeping the rate of viral spread ($b_i$) and infected cell death rate ($d_i$) at their default values (A) or in a range (B). 10,000 simulations were run for each panel, and each point corresponds to one simulation. With the exception of the investigated parameters, all parameters were at their default values.
(TIF)

**S3 Fig. Effect of birth and death rate of stromal cells on the outcome of virotherapy.** Therapeutic outcomes in relation to the birth ($b_s$) and death rates of stromal cells ($b_s$) for the Voronoi model were considered by keeping the rates of viral spread ($b_i$) and death ($d_i$) at their default values (A) or in a range (B). 10,000 simulations were run for each panel, and each point corresponds to one simulation. With the exception of the investigated parameters, all parameters were at their default values.
(TIF)

**S4 Fig. Effect of grid size and spatial configuration on the outcome of virotherapy.** Therapeutic outcomes in relation to the rates of viral spread via contact ($b_i$) and death rate of infected cells ($d_i$) for the three spatial configurations considered (regular 2D grid, 2D Voronoi model, regular 3D grid) and for various population sizes (A) about 500 lattice cells; (B) about 10,000 lattice cells; (C) about 64,000 lattice cells; (D) about 250,000 lattice cells. The white text in each panel indicates how the population size relates to the grid dimensions. 50,000 simulations were run for each panel, and each point corresponds to one simulation. With the exception of population size, all parameters were at their default values.
(TIF)

**S5 Fig. Effect of different forms of viral infection on the therapeutic outcome.** Simulation outcome in relation to the rates of viral spread via contact ($b_i$) and death rate of infected cells ($d_i$) in the 3D model for three different forms of viral infection. As indicated in the illustration on the left, virus infection in the tumour is initiated either in the centre (top row), or from the periphery (middle row), or in a random manner (bottom row). For each infection scenario 50,000 simulations were run, which were classified according to (A) their therapeutic outcomes; and (B) the number of the different types of cells at the end of the simulations. All parameters were at their default values. The colour code is based on the logarithm of cell numbers.
(TIF)

**S6 Fig. Comparison of our model with the results of Wodarz et al. (2012).** In their Figs 4 and 5, Wodarz and colleagues [14] compare the spatial dynamics of experimentally induced viral infections with the predictions of an agent-based model. Here, we illustrate that our

event-based model creates similar spatial patterns as the discrete-time Wodarz model. (A) In both models, virotherapy leads to the radial spread of the virus and the extinction of cancer cells if the viral infection rate exceeds the death rate of infected cells by a factor of at least 10. In the simulation shown, $b_i = 0.1$ and $d_i = 0.001$. (B) In both models, virotherapy results in the long-term coexistence of infected and uninfected cancer cells if the viral infection rate does neither exceed the death rate of infected cells nor the birth rate of uninfected cells by a factor of 10. In the simulation shown, $b_i = 0.1$, $b_c = 1.0$ and $d_i = 0.01$. In the experimental observations of Wodarz and colleagues, infected cells are labelled as green and non-infected cells as black. Regarding the simulations, we follow the conventions of Wodarz and colleagues to label infected cancer cells as red, non-infected cells as green, and empty space as white.
(TIF)

**S7 Fig. Therapeutic outcome for an increased parameter range of rate of viral spread via contact and death rate of infected cells.** In the graphs of the main text (Fig 3), the rate of virus spread ($b_i$) ranges from 0 to 5, while the death rate of infected cells ($d_i$) is from 0 to 2. For the three spatial configurations considered (regular 2D grid, 2D Voronoi model, regular 3D grid), the panels show the therapeutic outcome for a wider range of parameters, for two scenarios: (A) stromal cells cannot be infected by the virus; (B) stromal cells are sensitive to infection. Each panel in the figure is based on at least 10,000 simulations and each point represents one simulation. All parameters that are not varied are at their default values.
(TIF)

**S8 Fig. Consequences of sensitizing stromal cells for virotherapy.** (A) Therapeutic outcome in relation to the rates of viral spread via contact ($b_i$) and death rate of infected cells ($d_i$) for two spatial configurations (2D Voronoi model, regular 3D grid) and two different assumptions on stromal cells: stromal cells cannot be infected by the virus (top row); or stromal cells are sensitive to infection (bottom row). Each of the four panels represents 100,000 simulations. For the simulations in (A), the four panels indicate the number of stromal cells (B) and the number of resistant cancer cells (C) at the end of the simulation. The colour code is based on the absolute of cell numbers. (D) Therapeutic outcomes in the Voronoi model in relation to the rates of viral spread via contact and death rate of infected cells for different degrees of stromal cell susceptibility to viral infection. Each of the four panels represents 10,000 simulations.
(TIF)

**S9 Fig. Effect of the production rate of virus-resistant cancer cells on the simulation outcome.** (A) Therapeutic outcomes in the 2D Voronoi model in relation to the rates of viral spread via contact ($b_i$) and death rate of infected cells ($d_i$) for six probabilities of becoming resistant ($C_r$ ranging from 0 to $10^{-2}$ per cell division). Each panel represents 100,000 simulations, and each point corresponds to one simulation. (B) Numbers of different types of cells at the end of the simulation for the default value ($C_r = 10^{-5}$ per cell division). The colour code is based on the logarithm of cell numbers.
(TIF)

**S10 Fig. Effect of degree of resistance to viral infection.** (A) Therapeutic outcomes in the 2D Voronoi model in relation to the rates of viral spread via contact ($b_i$) and death rate of infected cells ($d_i$) for five degrees of resistance of cancer cells ($R_r$ ranging from 0 to 1) and two production rates ($C_r$ is $10^{-5}$ or $10^{-3}$ per cell division) of resistant cells. Each panel represents 10,000 simulations, and each point corresponds to one simulation. (B) The bar chart indicates the likelihood of the four outcomes for five values of degree of resistance ($R_r$) and the size of the four bars in the bar chart is proportional to the areas indicated by blue, yellow, red, and orange in (A). The bar chart for the production rate of resistant cells at $10^{-5}$ per cell division in (B) is

the same as depicted in Fig 4B.
(TIF)

**S11 Fig. Effect of time of treatment on the therapeutic outcome.** This figure corresponds to Fig 5 in the main text, which illustrates the effect of start of viral treatment ($T_i$) on the therapeutic outcome in the 2D Voronoi model. Here, the corresponding outcomes are shown in (A) for the regular 2D grid and 3D grid models. Effect of start of virotherapy ($T_i$) on the (B) number of stromal cells and (C) number of infection-resistant cancer cells at the end of the simulation is provided. Two scenarios are considered: stromal cells are resistant (top row) or sensitive (bottom row) to infection. The number of simulations per panel and the model parameters are as in Fig 5.
(TIF)

**S12 Fig. Effect of time of treatment with respect to the rate of viral spread and the death rate of infected cells on the therapeutic outcome.** (A) Effect of the time of treatment ($T_i$) on the therapeutic outcome in the Voronoi model and 3D model. For the same range of parameter values as in Fig 3 (rate of viral spread ($b_i$) and death rate of infected cells($d_i$)) 10,000 simulations were run and classified as to their therapeutic outcome. (B) The bar chart indicates the likelihood of the four outcomes for different values of $T_i$.
(TIF)

## Acknowledgments

We thank David Berg for sharing the source code of their computational model for our reference, Roger Chammas from the University of São Paulo for useful discussions and the Centre for Information Technology of the University of Groningen for their support and for providing access to the Peregrine High-Performance Computing cluster.

## Author Contributions

**Conceptualization:** Darshak Kartikey Bhatt, Thijs Janzen, Toos Daemen, Franz J. Weissing.

**Data curation:** Darshak Kartikey Bhatt.

**Formal analysis:** Darshak Kartikey Bhatt, Franz J. Weissing.

**Funding acquisition:** Franz J. Weissing.

**Investigation:** Darshak Kartikey Bhatt, Thijs Janzen, Franz J. Weissing.

**Methodology:** Darshak Kartikey Bhatt, Thijs Janzen, Toos Daemen, Franz J. Weissing.

**Project administration:** Franz J. Weissing.

**Resources:** Franz J. Weissing.

**Software:** Darshak Kartikey Bhatt, Thijs Janzen.

**Supervision:** Thijs Janzen, Toos Daemen, Franz J. Weissing.

**Validation:** Darshak Kartikey Bhatt, Thijs Janzen.

**Visualization:** Darshak Kartikey Bhatt, Thijs Janzen, Franz J. Weissing.

**Writing – original draft:** Darshak Kartikey Bhatt, Thijs Janzen, Toos Daemen, Franz J. Weissing.

**Writing – review & editing:** Darshak Kartikey Bhatt, Thijs Janzen, Toos Daemen, Franz J. Weissing.

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
