## [Decision Letter · Decision Letter 0]

27 Jun 2022

Dear Bhatt,

Thank you very much for submitting your manuscript "Modelling the spatial dynamics of oncolytic virotherapy in the presence of virus-resistant tumor cells" for consideration at PLOS Computational Biology.

As with all papers reviewed by the journal, your manuscript was reviewed by members of the editorial board and by several independent reviewers. In light of the reviews (below this email), we would like to invite the resubmission of a significantly-revised version that takes into account the reviewers' comments. As you will see, the reviewers are split - recommending Reject, Major Revision, Accept. A very important point that the reviewer who recommends Reject raises concerns novelty and innovation. I think it is really important to address this issue - are the results intuitively obvious or does the modelling add new biological insights? Can the model make new predictions that are counter-intuitive?

We cannot make any decision about publication until we have seen the revised manuscript and your response to the reviewers' comments. Your revised manuscript is also likely to be sent to reviewers for further evaluation.

Sincerely,

Philip K Maini

Associate Editor

PLOS Computational Biology

Feilim Mac Gabhann

Editor-in-Chief

PLOS Computational Biology

Reviewer's Responses to Questions

**Comments to the Authors:**

Reviewer #1: Dear Editor,

The article develops probabilistic, cell-based models to study the dynamics of anti-cancer virus therapy, known as oncolytic virotherapy. Their model considers four populations: normal healthy (stromal) cells, infected cells (cancerous or stromal), infection-sensitive cancer cells, and infection-resistant cancer cells.

Then, the authors conduct a systematic investigation of the model’s outcomes and how they depend on parameter values and other assumptions. They simulate three different types of cell-based models – 2D lattice, 3D lattice, and Voroni – and show that the results are qualitatively similar with a few variations.

A positive aspect of the work is that the authors develop and simulate three different types of models, 2D, 3D, and Voroni, which helps establish credibility that their results are not simply artifacts of the type of model.

Another positive aspect is that they conduct a systematic investigation of different scenarios and parameter combinations, such as the dependence of cancer eradication on the ratio of death rate of infected cells and viral spread rate; the influence of the timing of virotherapy; and influence of various parameters, e.g., production rate of resistant cancer cells, virus diffusion distance, and probability of virus infection. They also consider the possibility of allowing normal healthy cells to get infected and show that this scenario could increase the chance of complete cancer elimination, a successful outcome, but with the risk of increasing the population resistant cancer cells, leading to therapy failure.

The investigation could be even more systematic, such as by varying all model parameters and reporting variations and trends in outcomes as a table or histogram. This would make the study even stronger.

The main weakness of the study is that the results do not seem innovative or novel. They seem very sensible and often expected. In addition, the treatment suggestion to consider an anti-cancer virus that can also infect normal healthy cells in an attempt to eliminate cancer is also not a very new idea. Consequently, this article does not provide the level of novelty or innovation suitable for PLoS Computational Biology.

Minor comments:

In the abstract, in lines 17-18, replace

The model allows to investigate the dynamics of infection-sensitive and infection-resistant cells in tumor tissue in presence of the virus.

with

The model allows *us* to investigate the dynamics of infection-sensitive and infection-resistant cells in tumor tissue in *the* presence of the virus.

In line 138, it says

A virus-infected cell infects a neighboring node with a probability that is identical with the susceptibility of this node, where the susceptibility of a susceptible cancer cell is given by 1

Is it a reasonable assumption that cancer cells get infected by infected neighbors with probability 1? It probably will not change the results much to vary this assumption, but the authors ought to mention the possibility that the infection probability is less than 1 and briefly note how that affects outcome without needing to show additional simulation results. It seems that the authors vary this assumption in Figure 6, but they ought to mention here that they will do it so that it does not seem as if they are ignoring this possibility until the reader comes to Figure 6.

Around Table 1, the authors ought to provide a parameter estimates subsection to discuss the parameters and their estimates. Summarize why the estimated values are the way they are even if the values came from Berg et al. or another paper.

Also, it seems parameter b_i for the rate of viral spread via contact of infected with susceptible cells is not discussed anywhere in the text. The authors ought to discuss this parameter in the text, too.

In Figure 4B,C,D, maybe the authors should plot the fraction of resistant cancer cells. Is it possible that the 3D model has a higher number of resistant cells just because it has more cells altogether? Is the proportion similar though?

Reviewer #2: Overall feedback

In this manuscript, the authors develop a cell-based model of viral infection to investigate the dynamics of infection-resistance in tumour tissue. They compare three different variations of the model: two 2D models of tumour cell monolayers and a 3D model. They qualify how the therapeutic outcome is effected by properties of the virus, tumour and stromal cells as well as timing of treatment. Their simulations suggest three primary causes of therapy failure: rapid clearance of the virus, rapid selection of resistant cancer cells, and a low rate of viral spread due to the presence of infection-resistant healthy cells. Their model suggest that improved therapeutic efficacy can be achieved by sensitizing healthy stromal cells to infection. The results in this paper are very interesting and serve as a good study for the oncolytic virotherapy community. In places, however, I felt the assumptions of the model were not well supported or clarity around why certain avenues for investigation were ignored would be important to address. In turn, computational modelling of oncolytic virotherapy is a large field of research and I feel more justice could be done to reference previous models and their results in this area.

Major

• The authors introduce the notion of infection-resistant cancer cells impeding oncolytic virotherapy, however, this is not explained biologically and as such I feel gives rise to some confusion and questions. What are infection-resistant cancer cells? Why are they infection-resistant? Does the presence of infection-resistant cells depend on the virus, the genetic modification, and the underlying cancer line? Is the resistant due to a lack of conjugate receptor expression? Internal machinery needed for viral RNA replication? or is it resistance from IFN signalling? I believe it is very important that the authors outline what these infection-resistant cancer cells are to be sure that the readers can believe they exist as there is some debate in the field around infection-resistance cells in oncolytic virotherapy.

• Why do the authors conserve the number of cell spaces from 2D to 3D? Is this a reasonable assumption and are they sure this doesn’t impact their predictions? Wouldn’t the tumour need the same amount of space to grow in a 2D plane as it would if you were just considering the 3D tumour? I think some clarity around this might help and maybe a figure of a simulation in the 2D models and the 3D models and comparing them would help alleviate my concern here (Something like Figure 2, but showing for the same parameter values what the three model predictions look like?).

• Along the same vein as the previous comment the authors state “Supplementary figure 1 shows that the simulation outcome is only marginally affected by grid size” but unless I am understanding Figure 1 incorrectly, increasing the grid size gives rise to larger and larger regions of the parameter space denoting resistance cancer cell persistence and less of the parameter space giving rise to sensitive cancer cell persistence and eradication. This needs to be clarified. Perhaps this might also be an explanation for why there is a large different in Figure 4 between the 3D model and two 2D models?

• The authors state that “each node In the grid can be inhabited by one of four different cell types: healthy (stromal) cells, infected (cancer or stromal) cells..” shouldn’t infected cancer and infected stromal cells be considered differently? Surely infected cancer cells and infected stromal cells will have different properties for infecting neighbouring cells as they would replicate the virus differently? If the authors are assuming not then maybe this needs to be discussed and I wonder whether this assumption has any impact on the model predictions?

• Can the authors discuss whether it is a limiting assumption to assume virus can only be transmitted from an infected cell to a neighbouring cell point? Viruses can diffuse through the tumour after secretion from infected cells and so in theory could diffuse past neighbouring resistant cells and infect susceptible cells. Did the authors confirm what impact it might have on the model if the radius of infection was increased to neighbours twice removed? I know they introduced the assumption that dead cells can transmit further, but I’m curious as to whether if the assumption for dead cells was given to infected cells what impact this might have on the model predictions?

• It’s not surprising the authors did not find that infections in locations apart from the centre gave similar results as there was no spatial dependence on the likelihood of resistance, cell proliferation or viral replication. Could the authors comment on this? On whether the introduction of nutrient dependent (distance from edge) cell proliferation or incidence of infection-resistant mutation based on location to viral infection could impact the model predictions?

• Is it possible that the choice of tumour cell proliferation rate and other parameters like the death rate of healthy stromal cells could also be drivers of some of the sensitivity of the model predictions? Is it possible for the authors to comment on this or investigate this? It’s been shown that stromal density has an impact on OV efficacy, is this seen in your model? It might be part of varying stromal proliferation rate or density. In addition, could the authors elaborate on how one might make stromal cells “sensitive” to infection.

Minor

• Typo:

o “The model allows to…” – should be “allows us to”?

o Line 44 space in “research[11]”

o “See Fig. 6B for an illustration” do you mean 6A?

o “Figures 4BCD” maybe should be just “Figures 4B-D”?

• It feels like think there is a lack of review of the mathematical/computational literature surrounding agent-based models of oncolytic virotherapy, especially in the context of heterogeneous stromal tumours or Voronoi tessellations. Consider referencing the below

o https://doi.org/10.3390/cancers13215314

o doi: 10.1016/j.isci.2022.104395

o https://doi.org/10.1016/j.jtbi.2019.110052

• The lattice configuration in Figure 1A is unclear to me, I can see a green cell in the centre which I believe represents an infected cell, but then just a red region around it? Is this red region meant to represent 4 cells? Can this image be fixed to actually denote the lattice, similar to what is shown in the Voronoi Tessellation where there are lines for each cell?

• The authors should be very careful with using the terminology of “cells” to mean biological cells and “cells” to mean lattice points as it does get confusing in places. For example in supplementary figure 1 when the authors say “(a) 500 cells; (b) 10,000 cells” etc, do they mean initial number of cells or total cell lattice points? Maybe changing to lattice cells or lattice points will help?

• I found the description around S_s confusing. Is S_s between 0 and 1? Shouldn’t the susceptibility of stromal cells be less than or as great as cancer cells? How does this all relate to the rate b_i? Can the introduction of the infection probabilities be expanded upon and more detail given please (particularly making clear the assumptions around infection).

• Is Figure 2 the Voronoi 2D model or otherwise? Please state. What are the black regions in this figure? A suggestion would be to rotate Figure 2 90 degrees so we can see your animations clearer as they are very small currently.

• While the figures are lovely, I found the text on the majority of the figures way too small and difficult to read. Could the authors please consider fixing this for the readers?

• In Table 1 it says C_r is a production rate but in the text I believe it said it was a probability? Can this be clarified? Potentially the table needs a column saying the units so that it is clear what is a probability and what is a time-dependent rate?

• There is a difference between a “rate” and “probability” in the context of a cell-based model and the authors should elaborate on their explanation for the Gillespie algorithm. Consider adding a few steps of the algorithm to the supplementary information for reproducibility.

Reviewer #3: The review is uploaded as an attachment.

**Have the authors made all data and (if applicable) computational code underlying the findings in their manuscript fully available?**

Reviewer #1: Yes

Reviewer #2: Yes

Reviewer #3: Yes

PLOS authors have the option to publish the peer review history of their article (what does this mean?). If published, this will include your full peer review and any attached files.

Reviewer #1: No

Reviewer #2: No

Reviewer #3: No
---

## [Decision Letter · Decision Letter 1]

16 Nov 2022

Dear Bhatt,

We are pleased to inform you that your manuscript 'Modelling the spatial dynamics of oncolytic virotherapy in the presence of virus-resistant tumour cells' has been provisionally accepted for publication in PLOS Computational Biology.

Best regards,

Philip K Maini

Academic Editor

PLOS Computational Biology

Feilim Mac Gabhann

Editor-in-Chief

PLOS Computational Biology

Reviewer's Responses to Questions

**Comments to the Authors:**

Reviewer #1: The authors have sufficiently addressed by comments.

Reviewer #2: I am satisfied that the reviewers have responded to all of my comments

**Have the authors made all data and (if applicable) computational code underlying the findings in their manuscript fully available?**

Reviewer #1: Yes

Reviewer #2: None

PLOS authors have the option to publish the peer review history of their article (what does this mean?). If published, this will include your full peer review and any attached files.

Reviewer #1: No

Reviewer #2: No

---

## [Editor Report · Acceptance letter]

30 Nov 2022

PCOMPBIOL-D-22-00532R1 

Modelling the spatial dynamics of oncolytic virotherapy in the presence of virus-resistant tumour cells

Dear Dr Weissing,

I am pleased to inform you that your manuscript has been formally accepted for publication in PLOS Computational Biology. Your manuscript is now with our production department and you will be notified of the publication date in due course.

With kind regards,

Zsanett Szabo
